# Estimated Healthcare Costs of Melanoma and Keratinocyte Skin Cancers in Australia and Aotearoa New Zealand in 2021

**DOI:** 10.3390/ijerph19063178

**Published:** 2022-03-08

**Authors:** Louisa G. Gordon, William Leung, Richard Johns, Bronwen McNoe, Daniel Lindsay, Katharina M. D. Merollini, Thomas M. Elliott, Rachel E. Neale, Catherine M. Olsen, Nirmala Pandeya, David C. Whiteman

**Affiliations:** 1Population Health Department, QIMR Berghofer Medical Research Institute, Brisbane, QLD 4006, Australia; thomas.elliott@qimrberghofer.edu.au (T.M.E.); rachel.neale@qimrberghofer.edu.au (R.E.N.); catherine.olsen@qimrberghofer.edu.au (C.M.O.); nirmala.pandeya@qimrberghofer.edu.au (N.P.); david.whiteman@qimrberghofer.edu.au (D.C.W.); 2Cancer and Palliative Care Outcomes Centre and School of Nursing, Queensland University of Technology (QUT), Brisbane, QLD 4059, Australia; 3Faculty of Medicine, The University of Queensland, Brisbane, QLD 4006, Australia; d.lindsay@uq.edu.au; 4Wellington School of Medicine, University of Otago, Wellington 6242, New Zealand; william.leung@otago.ac.nz; 5Kenmore Skin Clinic, Moggill Rd, Brisbane, QLD 4069, Australia; richard.johns@hotmail.com; 6Social and Behavioural Research Unit, Department of Preventive and Social Medicine, University of Otago, Dunedin 9016, New Zealand; bronwen.mcnoe@otago.ac.nz; 7School of Health and Behavioural Sciences, University of the Sunshine Coast, Maroochydore, QLD 4558, Australia; kmerolli@usc.edu.au; 8Sunshine Coast Health Institute, Birtinya, QLD 4575, Australia

**Keywords:** melanoma, keratinocyte cancer, basal cell carcinoma, squamous cell carcinoma, cost-of-illness, Markov model, healthcare costs

## Abstract

Australia and Aotearoa New Zealand have the highest incidence of melanoma and KC in the world. We undertook a cost-of-illness analysis using Markov decision–analytic models separately for melanoma and keratinocyte skin cancer (KC) for each country. Using clinical pathways, the probabilities and unit costs of each health service and medicine for skin cancer management were applied. We estimated mean costs and 95% uncertainty intervals (95% UI) using Monte Carlo simulation. In Australia, the mean first-year costs of melanoma per patient ranged from AU$644 (95%UI: $642, $647) for melanoma *in situ* to AU$100,725 (95%UI: $84,288, $119,070) for unresectable stage III/IV disease. Australian-wide direct costs to the Government for newly diagnosed patients with melanoma were AU$397.9 m and AU$426.2 m for KCs, a total of AU$824.0 m. The mean costs per patient for melanoma ranged from NZ$1450 (95%UI: $1445, $1456) for melanoma *in situ* to NZ$77,828 (95%UI $62,525, $94,718) for unresectable stage III/IV disease. The estimated total cost to New Zealand in 2021 for new patients with melanoma was NZ$51.2 m, and for KCs, was NZ$129.4 m, with a total combined cost of NZ$180.5 m. These up-to-date national healthcare costs of melanoma and KC in Australia and New Zealand accentuate the savings potential of successful prevention strategies for skin cancer.

## 1. Introduction

Skin cancer is a major public health problem in many countries with fair-skinned populations. The two most common types of skin cancers are basal cell carcinoma (BCC) and squamous cell carcinoma (SCC), collectively called keratinocyte carcinoma (KC). Although less common, melanoma has much higher mortality than KC. Australia and New Zealand have the highest incidence of melanoma and KC in the world, attributed largely to the high ambient UV radiation to which Australians and New Zealanders are exposed [1], coupled with a high proportion of fair-skinned residents with European ancestry. Collectively, nearly 20,000 new cases of malignant melanoma are diagnosed in Australia [2] and New Zealand [3] annually. The incidence of KC is over 30 times higher than melanoma [4,5].

The high incidence of skin cancers, and the resources used in their management, translates to a high cost burden [6]. These resources range from general practitioner (GP) and specialist consultations, and include biopsies, ablations, excisions, skin grafts, imaging, lymph node sampling and removal, systemic pharmacotherapies, and follow-up care. For patients with more advanced stages of melanoma, new high-cost systemic pharmacotherapies have been approved for Government subsidy, as a part of Australia and New Zealand’s universal healthcare systems [7]. As melanoma case numbers are expected to increase in the foreseeable future, due mainly to population ageing, high-cost therapies will continue to strain healthcare budgets.

Studies identifying the magnitude of healthcare system costs are needed to understand the economic burden of skin cancer, for future healthcare and medical workforce planning, and for incorporating into future cost-effectiveness studies. Health system costs of skin cancer for Australia and New Zealand are out of date [8]. The last economic analysis undertaken in New Zealand is over a decade old [9], and this was based on data inputs from 1998. Much has changed since that time, and both the higher costs and higher number of cases of skin cancer means the overall financial burden is substantially increased. Australia and New Zealand face growing and ageing populations, increased overall incidence of skin cancers, higher use of new imaging technologies to improve detection, and new pharmacotherapies.

Particular challenges exist in determining the healthcare costs for skin cancer, and include data availability and coverage, decisions about including resources peripheral to confirmed cases (e.g., screening costs, benign skin neoplasms), or including costs of suspected skin cancers later found to be benign (case-finding costs). Previous cost-of-illness studies vary with their research approaches and decisions on capturing all possible resources involved [10,11,12,13].

To obtain a comprehensive and updated assessment of the health system costs of skin cancer in Australia and New Zealand, we used a modelling approach to estimate the national costs of diagnosing and treating skin cancers, and incorporated treatment modalities typically used in the two nations.

## 2. Materials and Methods

### 2.1. Overview

A cost-of-illness analysis was undertaken to provide current estimates of skin cancer costs in Australia and New Zealand using a granular pathways costing approach [14]. Four Markov decision-analytic models were developed; one each for melanoma and KC for each country. The key clinical pathways were created for each model, and the probabilities of diagnosis- and treatment-related services and medicines for skin cancer and associated unit costs were applied. We analysed patient-level data to obtain the probabilities of different resources used (e.g., GP and specialist consultations, biopsies, excisions, topical creams, medications, immunotherapies, etc.). Unit costs sourced from each country were attached to the defined pathways of care. The estimated costs of skin cancers included the additional resources used for managing suspected skin cancers, found to be benign on histopathology, and we accounted for multiple KCs or melanomas (new primaries) per person. The aggregated mean cost per person was computed, and, using current national skin cancer incidence rates, costs were extrapolated to the national level. Sensitivity and scenario analyses were performed to assess model uncertainty. The Consolidated Health Economic Evaluation Reporting Standards (CHEERS) statement guided this work [14].

### 2.2. Model Structures

Markov health state transition models were constructed separately for KC and melanoma in *TreeAge Pro 2021 R2* (TreeAge Software Inc., Williamstown, MA, USA), and included the major treatment pathways for KC and melanoma (Figure 1 and Figure 2). Models were analysed in annual cycles up to five years, when most resource use is expected. Face validity of the model pathways were assessed from clinical practice guidelines, from senior doctors working in skin cancer medicine, and from publications; with few exceptions, these were similar for both countries (Figure 1). Australian and New Zealand physicians jointly follow Australasian clinical practice guidelines, and receive training through medical speciality colleges. Both countries have universal healthcare systems, and their large caseload of patients with KCs are predominantly managed by GPs in primary care and dermatologists, whereas specialists (i.e., dermatologists, plastic surgeons, oncologists) are required for more difficult cases and advanced disease [15].

Melanoma Model: Melanoma diagnosis and treatment varied according to the severity or stage of disease (Figure 1). Surgical excision was the mainstay of treatment for *in situ* and stage IA melanoma. For stage IB and II, treatment included excision with or without sentinel lymph node biopsy (SLNB) plus observation. Those with positive nodes were upstaged to stage III pathways. Resectable stage III melanoma is excised, and may have complete lymph node dissection and adjuvant systemic therapies and/or radiotherapy. For advanced stage melanoma (stage IV), we modelled systemic therapies, including immunotherapies and targeted therapies for *BRAF* mutation-positive melanomas. Unlike Australia, the New Zealand Government currently does not subsidise adjuvant therapies for resectable stage III melanoma or ipilimumab or dabrafenib/trametinib for unresectable stage III or IV melanoma, and these were omitted from the New Zealand model. Follow-up skin examinations were assumed to occur bi-annually for up to 5 years, as per guidelines.

KC Model: Patients with suspected KC underwent a GP consultation involving a skin examination, and a proportion had a skin biopsy (Figure 2). The model randomly assigned a proportion of patients to receive a skin biopsy, and further assigned proportions to shave with curative intent or punch biopsy for pathology confirmation of malignancy. Patients were treated either by a GP or could be referred to a specialist. Treatments included surgical excision (predominantly), cautery and curettage, cryotherapy, or topical lotions. A small proportion were treated in hospitals with wide excision, radiotherapy (for perineural invasion), or Mohs micrographic surgery. A small proportion of patients also required a re-excision for unclear margins. Our primary model assumed annual follow-up skin examinations; in sensitivity analyses, we assumed bi-annually examinations for 5 years.

### 2.3. Data Sources

Model inputs came from various sources, but predominantly relied on health service use captured in the QSkin Sun and Health Study [16] (Appendix A). The QSkin study, located in Queensland, Australia, involves 43,794 participants recruited in 2010–2011, and who were aged 40–69 years. Participants were randomly selected from the Queensland Electoral Roll. The study population was 46% male (mean age of 57 years) and 54% female (mean age of 55 years). Most participants reported having white European ancestry (93%), and 59% had self-reported fair skin. The study population was comparable to the Queensland average with respect to education, employment, and body mass index [16]. Medicare claims data were linked to the QSkin cohort, and analysed to obtain frequencies of skin cancer medical services and medicines from 2010–2020.

Medicare is an Australian Government entity, and is part of a universal healthcare system that subsidizes most medical services for citizens via the Medical Benefits Scheme (MBS) and Pharmaceutical Benefits Scheme (PBS). The MBS includes all physician consultations, pathology, investigations, and procedures, whereas the PBS subsidises medicines and vaccines. For our primary analyses, we defined KC and melanoma cases as participants who received excision or non-excision codes or pharmaceutical items relating to skin cancer treatment (Appendix A). Separate item codes exist for melanoma versus KC. In total, this involved 16,514 persons treated using claims for KCs, and 1007 persons treated using claims for melanomas (including 109 persons treated with therapies for advanced-stage melanoma). In the absence of current New Zealand patient-level data, treatment probabilities from the QSkin cohort and other Australian estimates were used.

### 2.4. Probabilities

We analysed the PBS and MBS data for QSkin participants to obtain the probabilities for each service and medicine involved in skin cancer management using skin cancer-related item codes (Appendix A). We converted the probability of persons with multiple skin cancer treatments for KCs over 5 years (50.9%) to an annual probability using a rate-to-probability formula [17]. Surgical excision was the prominent treatment for KCs (90.5% after biopsy), and <3% of patients received imiquimod cream or radiotherapy (Appendix A). Excisions of suspected KCs that were subsequently diagnosed as benign lesions on histopathology were included.

Australian and New Zealand melanoma guidelines [18] recommend that patients with a melanoma greater than 1.0 mm (or >0.75 mm with other high-risk pathological features) be considered for SLNB for optimal staging and prognostic information. SLNB was included in the model at 60.4% [19] uptake (Appendix A); this may increase in the future with the updated guidelines.

There were only 109 cases with advanced melanoma in QSkin (as identified by PBS therapy codes), so we could not use these data to accurately estimate the probabilities of patients receiving pharmacotherapies for stage III or IV resectable and unresectable melanoma; instead, we used national PBS prescription records (Appendix A) [7]. Items were distinguished for first-line or adjuvant indications, and for initial or continuing treatment. We sourced survival time from international trials of each therapy (i.e., ipilimumab, nivolumab, pembrolizumab, dabrafenib/trametinib) [20,21,22,23,24,25]. For patients with fatal melanoma, the probability of moving to palliative care was calculated from the overall median 4- or 5-year survival rates of each therapy from pivotal trials, converted to annual probabilities (Appendix A).

### 2.5. Costs

The study took a healthcare cost perspective. This meant a predominantly Government perspective (State and Federal Governments) for Australia, and a mixed Government and patient perspective for New Zealand. In New Zealand, patients incur many costs for skin cancer treatment in the form of out-of-pocket expenses, but the proportions incurred by patients or Government are currently unknown. Australian mean costs for services and medicines were derived through analysis of the QSkin dataset. Where multiple items existed for a particular treatment (see Appendix A for excision items), we generated frequency-weighted mean costs. BCC and SCC costs were aggregated because this is how the Australian Government subsidises the diagnosis and treatment of these cancers via Medicare. Several New Zealand unit cost estimates were sourced from the New Zealand medicines schedule, PHARMAC, from hospital casemix reports and other published sources, but not all unit costs were available. In their absence, we applied the mean relative price ratio of known to unknown unit prices in Australia and New Zealand (1.91) (Appendix A).

Although most skin cancers are treated in community GP and specialist settings in Australia and New Zealand, hospitalisations, including same-day admissions, occur for KCs and melanoma. A hospitalisation rate of 44 per 10,000 patients with KC was used for treatments in hospitals [26]. We extracted the unit costs of SLNB, complete lymph node dissection, radiotherapy, and palliative care from national cost reports [27]. Hospitalisation costs for KCs and melanoma were extracted from two hospital costing datasets that included episode costs for ICD C43-44 [28,29]. We derived the costs of pharmacotherapy by analysing the treatment courses and duration for 109 QSkin participants treated with ipilimumab, nivolumab, pembrolizumab, and dabrafenib/trametinib. Among these participants, 89% of therapy scripts were claimed within 12 months of diagnosis, and the remainder was carried through to year 2 in the model. Probabilities of grade 3 or 4 adverse events were derived from trial evidence [20,21,22,23,24,25] (ranging from 15–28%) for systemic therapies, and hospital episode costs were applied.

### 2.6. Analyses

To address uncertainty in the model inputs, distributions were assigned around the mean values. We used beta distributions for probabilities, and gamma distributions for costs (Appendix A). We estimated mean costs and 95% uncertainty intervals (95% UI) using Monte Carlo simulation, re-sampling from the parameter distributions with 10,000 iterations. UIs were derived by ranking the costs, and extracting the 2.5th and 97.5th percentiles. In addition, to assess the key drivers of mean costs, one-way sensitivity analyses on each variable addressed how potential uncertainty of inputs would vary the main findings. In one-way sensitivity analyses, we used 10–20% margins of error for point estimates, or the 25th/75th percentiles for individual-level data. In scenario analyses, to estimate the additional costs in diagnosing melanoma, we re-ran the model using estimates of ‘number needed to biopsy’, 3.5 for specialists, and 14.6 for GPs [30,31]; applied as multipliers to biopsy costs in the model.

We estimated nationwide costs by multiplying the mean cost per patient with the latest annual incidence of KC [4,5] and melanoma [2]. It is mandatory to report melanoma diagnoses to Australian State and New Zealand cancer registries. However, although melanoma *in situ* is routinely notified to cancer registries, it is not always reported in publications. Since they still incur healthcare resources, a 1:1 incidence rate ratio of *in situ* to invasive melanoma was assumed; this is higher than previous reports, but allows for the upward trend in the incidence of *in situ* relative to invasive [32]. To calculate the costs from 2021–2025, the incremental mean costs from the models were produced for years 2–5, capturing the multiplicity of skin cancers for some people, as well as the ongoing surveillance. These were combined with the mean costs (year 1) for first incident cases. Costs were presented in 2021 Australian (AU$) and New Zealand dollars (NZ$) exclusive of the goods and services tax.

## 3. Results

In Australia, the mean cost of melanoma per patient (all stages) was AU$11,787 (95%UI: $9128, $14,921), and ranged from AU$644 (95%UI: $642, $647) for melanoma *in situ* to AU$100,725 (95%UI: $84,288, $119,070) for unresectable stage III/IV disease (Figure 3). The mean cost of melanoma (all stages) was most sensitive to variation in the cost of nivolumab therapy, probabilities of resectable and unresectable stage III melanomas, and the costs of other high-cost therapies (Appendix A). The total cost to Australia for newly diagnosed patients with melanoma for 2021 was AU$397.9 m (Table 1), with half of this for melanoma *in situ* (AU$198.9 m) (Figure 4).

For KCs, the mean cost per patient was AU$525 (95%UI: $452, $655). KC cost was most sensitive to the costs of excisions, skin grafts and flaps, pathology and biopsy, and the probability of skin graft or flap and having multiple KCs within 12 months (Appendix A). The total cost to Australia in 2021 for new patients with KCs was AU$426.2 m (Table 1). For melanoma and KC combined, the total costs were highest in Queensland, followed closely by New South Wales, both states making up 61.4% of the national cost burden.

In New Zealand, mean costs for melanoma (all stages) were NZ$8001 (95%UI: $6748, $9454), ranging from NZ$1450 (95%UI: $1445, $1456) for melanoma *in situ* to NZ$77,828 (95%UI $62,525, $94,718) for unresectable stage III/IV disease (Figure 3). The mean costs of melanoma were sensitive to the probability of stage III resectable or unresectable melanoma, and the cost of interferon therapy, and the cost of nivolumab and pembrolizumab (Appendix A). The total melanoma costs for new patients in 2021 was NZ$51.2 m (Table 1), with half of this comprising melanoma *in situ* (NZ$25.6 m) (Figure 4).

For KCs, the mean patient cost was NZ$1167 (95%UI: $1024, $1431). KC cost was most sensitive to the cost of a GP skin examination and costs of excisions, pathology, and skin grafts (Appendix A). The total cost to New Zealand in 2021 for new patients with KCs was NZ$129.4 m, and the total combined cost of melanoma and KCs was NZ$180.5 m (Table 1).

Most diagnosis and treatment costs for melanoma are resolved in a one-year period, and additional surveillance costs continue over time. However, the high multiplicity of KCs plus persons from 2021 who have multiple KCs in subsequent years meant overall costs rose to AU$1.2 billion and NZ$295.3 m by 2025 (Table 2). Although the mean costs per KC were much lower than for melanoma (all stages), over a 5-year period, first incident and subsequent episodes of KCs for patients produced similar healthcare costs than patients with melanoma in Australia (Table 2), and were 2-fold higher in New Zealand.

Scenario analyses to test for variation in KC incidence and the ‘number needed to biopsy’ showed large differences in the base findings. When the ‘number needed to biopsy’ was set at 3.5 for specialists, and 14.6 for GPs, Australian melanoma costs would rise to AU$15,011 per patient, equating the total cost to diagnose and treat melanoma as AU$506.7 m. The corresponding costs for melanoma in New Zealand were NZ$9537 per patient, with a total cost of NZ$60.1 m. If the estimated incidence of KCs were 20% higher for Australia and New Zealand, the costs for KCs would be AU$511.3 m and NZ$226.2 m, respectively.

## 4. Discussion

We found that each year, diagnosing and treating people newly diagnosed with skin cancer resulted in direct healthcare costs of AU$824.0 m in Australia, and NZ$180.5 m in New Zealand. Costs will be substantially higher in year 5, where people have repeated skin cancers excised each year. In Australia, the costs of melanoma were similar to those for KC in the first year, but substantially lower for New Zealand. Taking into account first incident and subsequent episodes of skin cancers over 5 years, the healthcare cost burden for KCs exceeds that for melanomas at year 5 in Australia and New Zealand, with the gap between KC and melanoma costs narrowing more in Australia due to subsidised access to novel melanoma therapies.

In Australia and New Zealand, health expenditure on KC and melanoma are the highest in the world relative to population size [6]. However, new advances in melanoma treatments have meant past estimates of melanoma treatment costs are outdated [6]. It is difficult to estimate the costs of disease when many data sources are required and/or are lacking. In the absence of current New Zealand patient-level data, treatment probabilities from the QSkin cohort were used. Previous New Zealand estimates also used Australian data [9]. In a 2009 New Zealand report, the New Zealand Government costs of melanoma were estimated at $5.7 m and $51.4 m for KCs (total $57.1 m) [9]. O’Dea concluded these costs were likely to underestimate the true cost due to potential underreporting of KCs and outdated inputs. Two reports have since estimated the New Zealand health system cost of melanoma: one at NZ$18.1 m [33], and another at NZ$24.3 m [34]. These estimates contrast with the $129.4 m in this study for KCs, and $51.2 m for melanoma. Though our estimates are substantially higher, they include current incidence rates, new high-cost pharmacotherapies, and melanoma in situ.

In the latest Government report on cancer expenditure in Australia, KC ranked second-highest after ‘other benign, *in situ* and uncertain neoplasms’ at $1.3 billion in 2018–2019 [35], whereas melanoma ranked 9th on $358 m (total for melanoma and KC, $1.7 billion). These costs included all hospital, primary care, and pharmaceutical costs, included patient co-payments (approximately 16.3% of total expenditure [36]), and both incident and prevalent skin cancer. A combined top-down and bottom-up approach was used. In the absence of patient co-payment contributions, we estimate the costs to be $1.4 billion. This is higher than the $1.2 billion for incident and subsequent episodes of skin cancer we estimated, most likely due to the different methods used and driven by the uncertainty in KC incidence data. We previously estimated the cost of managing incident melanoma cases at $187–216 m per year in 2017 [8]. Those earlier estimates excluded several pharmacotherapies for advanced melanoma or therapies that have recently been approved for use in resectable stage III disease as adjuvant treatment; this change in the availability of subsidised therapies explains the higher estimate reported here (AU$397.9 m).

Pathways or bottom-up costing approaches are beneficial when disease pathways and unit costs are known, and can overcome some of the issues with top-down approaches, which require attribution of broad generic costs to a particular disease, although no method is superior [35]. Our micro-costing bottom-up approach allowed us to precisely aggregate the major components of the care pathway, and assess key drivers. We used data from a large sample of individuals linked to the Queensland Cancer Registry and the Government’s universal health insurance dataset. Our analysis also accounted for the proportion of people having multiple skin cancers, and their subsequent treatments. We performed sensitivity analyses around input variables where some uncertainty exists. 

The proportion of patients at each melanoma stage was a critical variable in the model, and we used estimates from a large patient series that included US, European, and Australian cases [19] with stage IA to III patient proportions. However, this was supplemented by an estimate of stage IV probability (3.6%) from the NSW Cancer Institute, and an assumption of the ratio of invasive to *in situ* melanoma. Since the treatments and costs are distinctly different by stage, with the cost of advanced stage disease 100-fold higher than for early-stage disease, accurate stage distributions are important for cost estimation. Accurately distinguishing between skin cancers and benign lesions during skin examination, and potentially avoiding treatment of benign lesions, is also an important factor in the overall economic burden. 

Our study has several limitations. The QSkin Study sampled exclusively from the population of Queensland, and it is unclear if the treatment experiences of Queensland patients with KC and melanoma reflect those of the wider Australian and New Zealand populations. In Australia and New Zealand, it is not mandatory for KCs to be notified in cancer registries as they are for all other cancers, and therefore, relying on other proxies for incidence may be problematic for cost estimation. Our model was limited in scope, and did not include all skin cancers, for example, Merkel cell carcinoma, metastatic KCs, and other rare skin cancers. Medicare claims data is a rich data source, but relying on this administrative dataset misses clinical data necessary to explain resource use. It is particularly difficult to disentangle whether, for example, a person with multiple excisions is experiencing a new KC, or having a previously excised lesion re-excised. A further challenge is understanding the multiple settings involved in skin cancer medicine, and who exactly incurs the cost—private or Government providers. Further research on these aspects, and using primary care datasets with clinical encounter and patient diagnosis information, would be valuable. Finally, though we aimed to focus on Government expenditure, the inclusion of patient medical out-of-pocket expenses cannot be ignored for Australia or New Zealand. The healthcare costs are conservative because we have not captured indirect costs related to carers attending to patients, lost income, unfunded anticancer drugs, as well as other intangible costs associated with discomfort, scarring, anxiety, or pain from treatments [37]. In particular, patients requiring serial treatments and ongoing surveillance are likely to have impaired quality of life [37], and recurrent medical expenses.

### Implications

Our study highlights the potential cost-savings to Government for encouraging individuals to protect their skin, and reduce these largely preventable cancers. In doing so, we have provided the economic case to support public health skin cancer prevention programs and policies. Unlike other cancers, the dominant cause of skin cancer is exposure to ultraviolet radiation (UVR), which accounts for almost all KCs, and over 80% of melanoma [38]. Since exposure to UVR can be prevented, a large proportion of skin cancers and their associated economic burden can also be prevented by avoiding UVR exposure through sun protection measures [38]. Despite some reports that melanoma incidence in younger adults is declining [39], healthcare costs of skin cancer are expected to rise further due to the joint effects of health price inflation, new technologies and medicines, and ageing demographics [36,40]. Skin cancer prevention campaigns require financial commitment by Governments and interest groups. Currently, the dermatology workforce in both Australia and New Zealand is under significant pressure to manage the vast numbers of new and existing patients with skin cancers [41,42]. Given that skin cancer prevention campaigns are effective [43] and cost-effective in both the short and long term [6,44,45], it is critical that greater investment in prevention occurs to ensure more sustainable and efficient health systems.

## 5. Conclusions

In summary, we have reported contemporary costs of treating melanoma and KC in Australia and New Zealand. Cost-of-illness studies are important for understanding the economic impact of disease, and, with a largely preventable disease such as skin cancer, the findings will be useful for future resource planning, for cost-effectiveness studies of new skin cancer interventions, and for analysing the potential impact of prevention and early melanoma detection strategies.

## Figures and Tables

**Figure 1 ijerph-19-03178-f001:**
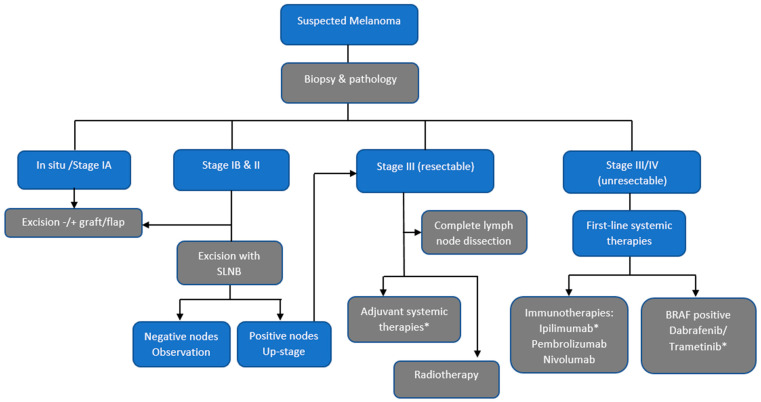
Main pathways for treatment of melanoma. * Adjuvant systemic therapies for resectable stage III melanoma or ipilimumab, dabrafenib/trametinib for unresectable stage III or IV melanoma are not subsidised in NZ, and these were omitted from the NZ model, and interferon therapy was added. SLNB = sentinel lymph node biopsy.

**Figure 2 ijerph-19-03178-f002:**
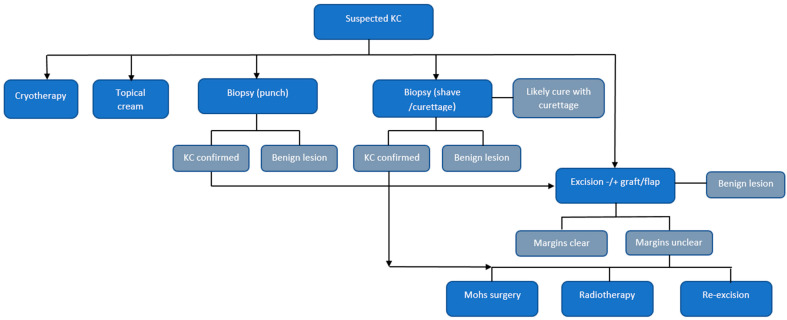
**Main pathways for treatment of keratinocyte cancer.** BCC/SCC have not been separated nor categorised by invasive, superficial, etc., because treatment is largely the same. Excision is the mainstay treatment modality. Radiotherapy is required if perineural invasion has occurred, performed in hospital, or for field treatment of multiple SCCs in the same area, e.g., forehead. Mohs surgery can be performed for an improved cosmetic outcome in a difficult facial area. Topical creams include imiquimod and 5-FU fluorouracil. GPs and skin cancer GPs perform much of the excisional treatments, whereas dermatologists and plastic surgeons treat more complex cases. A small proportion are referred to hospital dermatology departments where organ transplant recipients (as a very high-risk group) are also treated for skin cancers.

**Figure 3 ijerph-19-03178-f003:**
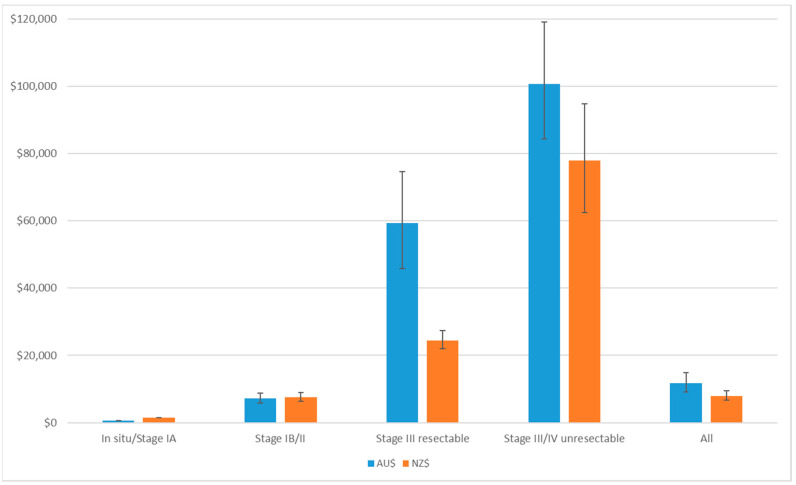
First-year cost of melanoma per patient by stage in Australia (AU$) and NZ (NZ$). Error bars are the uncertainty interval generated from 10,000 Monte Carlo simulations, and 2.5% and 97.5% percentiles.

**Figure 4 ijerph-19-03178-f004:**
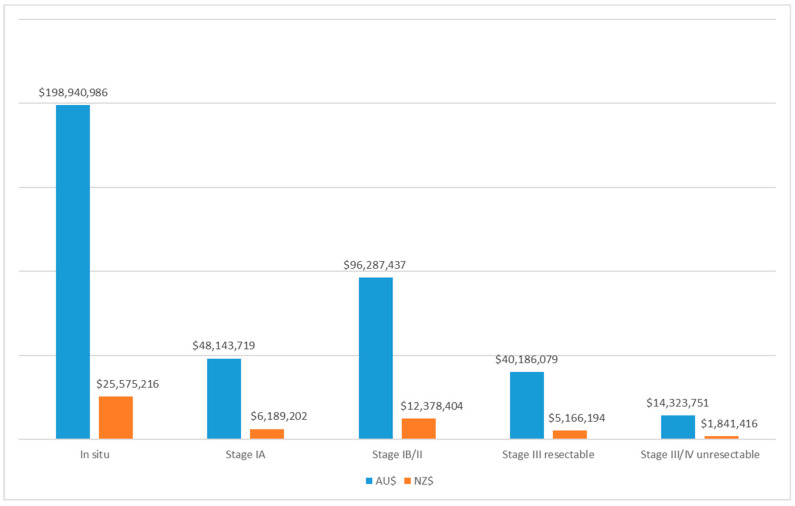
Total cost of first incident melanomas in 2021 by stage in Australia (AU$) and NZ (NZ$).

**Table 1 ijerph-19-03178-t001:** Estimated costs of new persons with skin cancers in Australia and New Zealand, 2021.

	NSW	VIC	QLD	WA	SA	TAS	NT	ACT	AUST. ($AU)	NZ ($NZ)
Est. 2021 population (million)	8.2	6.7	5.2	1.8	2.7	0.5	0.2	0.4	25.7	5.1
**Melanoma**										
Melanoma incidence ^1^ per 100,000 persons	61.1	51.0	81.1	62.0	46.4	58.5	54.7	56.5	62.4	62.4
Est. no. persons with invasive melanoma	5000	3404	4222	1100	1242	317	135	244	16,878	3197
Est. no. persons with *in situ* melanoma	5000	3404	4222	1100	1242	317	135	244	16,878	3197
Annual mean cost per melanoma (all stages)	$11,787									$8001
Total cost for melanoma (million)	$117.9	$80.2	$99.5	$25.9	$29.3	$7.5	$3.2	$5.8	$397.9	$51.2
**Keratinocyte cancers**										
Incidence of KC per 100,000 persons (lesion-based) ^2^	2799	1638	6174	2055	2620	133	174	148	3154	2165
Estimated no. persons with KC ^3^ (1000)	229.2	109.3	321.4	36.5	70.1	0.722	0.429	0.639	812.1	110.9
Mean cost per KC to Aust/NZ Govt	$525									$1167
Total cost for KC to Medicare (million)	$120.3	$57.4	$168.7	$19.1	$36.8	$0.4	$0.2	$0.3	$426.2	$129.4
**Total annual cost of skin cancer (million)**	$238.2	$137.6	$268.2	$45.1	$66.1	$7.9	$3.4	$6.1	$824.0	$180.5

^1^ Invasive melanoma, estimated counts (16,878), and age-adjusted incidence (62.4) in Australia in 2021, predicted from AIHW 2021 Cancer Data in Australia. State numbers are estimated. NZ assumed same rate. ^2^ NZ estimate is crude rate per 100,000 in 2013 population (Sneyd 2018), and Aust. estimates are age-standardised per 100,000 to Aust. Std population (Pandeya 2017). ^3^ Both are lesion-based incidence rates. Estimated persons with KC in NZ in 2018 was 90,400, but this was stated to be an underestimate (Sneyd 2018). Estimated persons with KC in Australia in the above table equates to 3% prevalence of all persons in a given year. Notes: Population size: estimates were from Australian Bureau Statistics (17/6/21) with 0.3% growth each year, and NZStats with 0.64% growth each year. Melanoma incidence: Australian Institute of Health and Welfare (AIHW) 2021 Cancer Data in Australia; Canberra: AIHW (https://www.aihw.gov.au/reports/cancer/cancer-data-in-australia accessed on 18 November 2021) by state and projections for 2021; assumed the same for NZ based on IARC Global Cancer Observatory with same rates for NZ and Australia. No. of cases in NZ in 2015 was ~2500. Estimated *in situ* cases: 50% of all melanoma cases or 1:1 incidence rate ratio. KC incidence: Australia from Pandeya et al. (2017) Med J Aust.; NZ from Sneyd, M.J., and Gray, A. (2018). *Expected non-melanoma skin (keratinocytic) cancer incidence in New Zealand for 2018.* Wellington: Health Promotion Agency. Average costs per melanoma and per KC: Calculated from Australia and NZ modelling of pathways of care in this study.

**Table 2 ijerph-19-03178-t002:** Costs of skin cancers in Australia and New Zealand projected over 5 years for new cohorts from 2021. ($ million).

	2021	2022	2023	2024	2025
**Australia**					
**Melanoma**					
First incident cases of melanoma ^1^ (person-based)	33,756	34,063	34,373	34,686	35,001
Continuing cohorts (minus deaths)	0	32,441	63,884	94,395	124,067
Cost of first incident cases	$397.9	$409.6	$413.3	$417.1	$420.9
Cost of subsequent episodes of melanoma	$0	$16.1	$28.2	$37.9	$45.9
Total melanoma cost	$397.9	$425.7	$441.6	$455.1	$466.8
**KC**					
First incident cases of KCs (person-based)	812,103	819,491	826,946	834,470	842,061
Continuing cohorts (minus deaths)	0	811,343	1,629,284	2,453,858	3,285,103
Cost of first incident cases	$426.2	$438.7	$442.7	$446.7	$450.8
Cost of subsequent episodes of KCs	$0	$66.6	$131.0	$192.0	$250.0
Total KC cost	$426.2	$505.3	$573.7	$638.7	$700.8
**Total Melanoma and KC cost (AU$)**	$824.0	$931.0	$1015.3	$1093.8	$1167.6
**New Zealand**					
**Melanoma**					
First incident cases of melanoma (person-based)	6393	6451	6510	6569	6629
Continuing cohorts (minus deaths)	0	6031	11,758	17,182	22,303
Cost of first incident cases	$51.2	$51.6	$52.1	$52.6	$53.0
Cost of subsequent episodes of melanoma	$0	$5.2	$9.7	$13.9	$17.6
Total melanoma cost	$51.2	$56.8	$61.8	$66.5	$70.7
**KC**					
First incident cases of KCs (person-based)	110,884	111,893	112,911	113,938	114,974
Continuing cohorts (minus deaths)	0	110,727	222,305	334,745	448,055
Cost of first incident cases	$129.4	$130.5	$131.7	$132.9	$134.1
Cost of subsequent episodes of KCs	$0	$0.111	$0.222	$0.335	$0.448
Total KC cost	$129.4	$154.6	$179.1	$202.4	$224.7
**Total Melanoma and KC cost (NZ$)**	$180.5	$211.4	$240.9	$268.9	$295.3

^1^ Includes both invasive and *in situ* cases of melanoma. Data sources: Mortality(AIHW 2021)-projected deaths are declining slightly since 2017, incremental costs per person from modelling in this study for each subsequent year, annual increase in new cases of melanoma 1.0091 (AIHW) applied to both melanoma and KCs (equivalent to 2.04 percentage points from 55.3 to 57.34 age-adjusted rate from 2021 to 2025), static population growth over next 5 years (minimal natural growth only due to COVID-19 border restrictions and no immigration, health inflation included (AIHW average 2.02% each year)).

## Data Availability

Restrictions apply to the availability of these data. Data was obtained from Services Australia, and analyses of these data are available from the authors with the permission of Services Australia.

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
