# Peer review of "Estimated Healthcare Costs of Melanoma and Keratinocyte Skin Cancers in Australia and Aotearoa New Zealand in 2021"

_ijerph, 2022, doi:10.3390/ijerph19063178_

Round 1
Reviewer 1 Report
This study used a cost of illness analysis using the Markov decision analytics model to estimate the average cost of health service and medicine for skin cancer management. This is exciting research on an important topic. However, a few things need to be taken care of before accepting for publication. The main issue with this manuscript is that it needs formatting. The quality of the presentation is poor. Some tables can be moved to Appendix.
The introduction is too short and lacks motivation for research.
Please add policy implications and limitations of this research.
Please follow journal guidelines for references.
Reviewer 2 Report
Very well written paper on a topic that hasn’t received enough attention, a pleasure to read! I only have a few very minor comments.
- The last sentence on pg 3 states: Medicare claims data were analysed to obtain frequencies of skin cancer medical services and medicines from 2010-2020. Was the Medicare data linked to the QSkin cohort?
- Small typo on pg 5: We extracted the unit costs of SLNB, complete lymph node dissection, radiotherapy, and palliative care was extracted from national cost reports
- In Table 1, can the authors please clarify what the column “estimates’ refers to.
